# Diversity and Biological Characteristics of Seed-Borne Bacteria of *Achnatherum splendens*

**DOI:** 10.3390/microorganisms12020339

**Published:** 2024-02-06

**Authors:** Jie Yang, Jinjing Xie, Haiyan Chen, Shaowei Zhu, Xuan Hou, Zhenfen Zhang

**Affiliations:** Key Laboratory of Grassland Ecosystem, Ministry of Education, Pratacultural College, Gansu Agricultural University, Lanzhou 730070, China; yangjie@st.gsau.edu.cn (J.Y.); xiejj@st.gsau.edu.cn (J.X.); chenhy@st.gsau.edu.cn (H.C.); zhusw@st.gsau.edu.cn (S.Z.); houx@st.gsau.edu.cn (X.H.)

**Keywords:** *Achnatherum splendens*, seed-borne bacterial diversity, motility, biofilm, antibiotic resistance

## Abstract

As a high-quality plant resource for ecological restoration, *Achnatherum splendens* has strong adaptability and wide distribution. It is a constructive species of alkaline grassland in Northwest China. The close relationship between seed-borne bacteria and seeds causes a specific co-evolutionary effect which can enhance the tolerance of plants under various stresses. In this study, 272 bacterial isolates were isolated from the seeds of *Achnatherum splendens* in 6 different provinces of China. In total, 41 dominant strains were identified, and their motility, biofilm formation ability and antibiotic resistance were analyzed. The results showed that the bacteria of *Achnatherum splendens* belonged to 3 phyla and 14 genera, of which Firmicutes was the dominant phylum and Bacillus was the dominant genus. The motility and biofilm formation ability of the isolated strains were studied. It was found that there were six strains with a moving diameter greater than 8 cm. There were 16 strains with strong biofilm formation ability, among which Bacillus with biofilm formation ability was the most common, accounting for 37.5%. The analysis of antibiotic resistance showed that sulfonamides had stronger antibacterial ability to strains. Correlation analysis showed that the resistance of strains to aminoglycosides (kanamycin, amikacin, and gentamicin) was significantly positively correlated with their biofilm formation ability. This study provides fungal resources for improving the tolerance of plants under different stresses. In addition, this is the first report on the biological characteristics of bacteria in *Achnatherum splendens*.

## 1. Introduction

*A. splendens* is a perennial tussock herb belonging to the *Achnatherum* genus of Gramineae. It is widely distributed in China and grows as a constructive species in the arid grassland of Northwest China [1,2]. As the dominant population of temperate grassland, *A. splendens* can be used as winter and spring forage by livestock. *A. splendens* growing along the Qinghai-Tibet Railway can not only prevent wind and sand and stabilize embankment, but also plays a good role in soil and water conservation. Seeds are essential throughout the life cycle of plants, which can exist in the environment for a long time in a dormant state until the growth conditions are suitable and then grow into complete plants [3]. Species-borne bacteria refer to all bacteria that parasitize inside and on the surface of seeds, and their colonization plays different roles in plant growth and health at different growth stages [4,5]. Seed-borne bacteria can exist in the whole life cycle of seeds, including seed germination and development to growth results and post-harvest storage and processing. It can also ensure the growth and reproduction of seed-borne bacteria in the next generation through vertical transmission and rhizosphere colonization. It is precisely because seed and seed-borne bacteria affect seed germination and plant growth in plant-bacterial interactions and other forms [6] that it is that great significance to clarify the diversity of seed-borne bacteria. However, *A. splendens* seed-borne bacteria are inevitably stressed by antibiotics in the environment due to the food chain circulation in the environment. Therefore, it is of great significance to explore the diversity and characteristics of *A. splendens* seed-borne bacteria to alleviate the stress of plants under adversity and the further utilization of seed-borne bacteria.

At present, antibiotic resistance has received extensive attention as one of the global problems that threaten human health. In 2019, researchers found that nearly 6.2 million people died directly or indirectly from antibiotic resistance [7]. Since the discovery and application of penicillin, the antibiotic industry has risen rapidly [8]. It is widely used in medical treatment, agriculture and animal husbandry. Avermectins (AVMs) are commonly used to control grasshoppers and caterpillars in grasslands [9,10]. Livestock in Northwest China rely on grassland-based natural vegetation, and the use of antibiotics is limited to livestock disease management. Although the dose of antibiotics is significantly lower than that of CAFOs, it can still cause drug resistance in animal and environmental reservoirs [11,12,13].

The extensive use of antibiotics leads to bacterial resistance, which increases the number of multi-antibiotic resistant bacteria, making common infections unable to be effectively treated [14]. The food chain [15], the horizontal spread of antibiotic resistance genes (ARGs), inadequate sanitary conditions, population mobility, and trade in live animals and meat make antibiotic resistance spread throughout the ecosystem [16]. In large-scale studies, it has been found that there is a significant correlation between antibiotic use and antibiotic resistance in livestock and human populations [17,18]. One of the reasons for bacterial resistance is the formation of biofilms. Biofilm has certain resistance to fungicides such as antibiotics and harsh environments, which can help bacteria better adapt to the living environment, and bacterial motility is crucial to the adhesion period of biofilm formation [19,20,21]. Different regions have bred different microbial resources due to differences in climatic conditions and geographical locations. As a reproductive organ of forage grass, the bacteria carried by seeds can not only transmit them by vertical transmission [22], but also bring them out by seed dispersal. If the bacteria carry the antibiotic resistance gene, the gene will also spread, and the antibiotics deposited in the environment will accumulate.

Therefore, in this study, *A. splendens* species from Xinjiang Uygur Autonomous Region, Inner Mongolia Autonomous Region, Ningxia Hui Autonomous Region, Qinghai Province, Gansu Province, Jiangsu Province and other places were used as research materials. Through the isolation, purification, phenotypic and molecular identification of the species-borne bacteria, the bacterial motility, biofilm formation ability and antibiotic resistance were determined, and the relationship between the three was analyzed in order to preliminarily determine the diversity of culturable bacteria in *A. splendens* and to clarify the biological characteristics of bacteria in *A. splendens*. Screening out strains with strong resistance and applying them to plants under antibiotic pollution, observing whether bacteria can improve the sensitivity of plants to antibiotics and slow down the persecution of antibiotics on plants, can provide a theoretical basis for the biological means to control antibiotic pollution, which is also our follow-up work.

## 2. Materials and Methods

### 2.1. Experimental Materials

The grass species used in the experiment and the grass species related information are shown in Table 1.

Reagents and medium used in the experiment: TSA medium, TSB medium, swimming medium, crystal violet staining solution.

### 2.2. Methods

#### 2.2.1. Isolation of Bacteria

The prepared TSA medium was placed in the incubator for 1 day, and used after confirming the medium was pollution-free. A total of 1 g of wild *A. splendens* from 6 different regions were weighed and placed in sterile pots. The disinfection treatment was first disinfected with 75% alcohol for 2 min, and then disinfected with 1% NaClO for 3 min. After disinfection, it was washed four times with sterile water to remove the disinfectant on the surface. Then 10 mL sterile water was added to the disinfected seeds for full grinding. After standing for 10 min, the diluent was prepared: 100 μL supernatant was taken and diluted 10 times in 900 μL sterile water to obtain 10^−1^, 10^−2^, 10^−3^, 10^−4^, 10^−5^ diluents. An amount of 100 μL was evenly coated on the TSA medium and cultured in darkness at 28 ± 1 °C in a constant temperature incubator. After 3 days, the single colony was purified and numbered, and then the strain selected in the TSB medium was supplemented with 15% glycerol and stored at −80 °C.

#### 2.2.2. 16S rDNA Molecular Identification of Bacteria

A TIANamp Bacteria DNA Kit (Tiangen Biochemical Technology (Beijing) Co., Ltd., Beijing, China) bacterial genomic DNA extraction kit was used to extract the DNA from the isolated and purified bacteria. The operation steps were carried out according to the kit instructions.

After the DNA extraction, PCR amplification was performed. The amplified primer sequences were: ① forward primer 27F: 5′-AGAGAG TTTGATCMTGGCTCAG-3′, ② reverse primer 1492R: 5′-GGYTACCTTGTTA CGACTT-3′. Among them, 1492R is a universal primer and 27F is a bacterial specific primer. The combination of 2 primers can effectively amplify the 16S rDNA of all bacterial groups. PCR amplification procedure: ① pre-denaturation at 95 °C for 4 min; ② cycle: 94 °C, 1 min; 50 °C, 1 min; 72 °C, 2 min, 34 cycle; ③ repair extension: 72 °C, 10 min; ④ preservation: 4 °C. After PCR amplification, the obtained products were sent to Shanghai Parsons Biotechnology Co., Ltd. (Shanghai, China) for sequencing. The sequences were aligned using BLAST in the NCBI database (http://www.ncbi.nlm.ni-h-gov/blast.cgi, (22 November 2023)). The phylogenetic tree was constructed by using the software MEGA 11.0 with the highest homologous strain sequence as the reference object.

#### 2.2.3. Determination of Swimming Motility

According to the method of Alías-Villegas [23], the bacteria were picked up with sterile toothpicks and cultured in a swimming plate at 28 °C for 20 h. The diameter of the turbid area formed by the diffusion of the bacteria was observed and measured.

#### 2.2.4. Determination of Biofilm Formation Ability

According to the method of Stepanovich S., [24] the isolated and purified strains were inoculated on fresh TSA plates for 24 h, and the strains were selected and cultured in a sterilized TSB medium overnight. The concentration of the bacterial solution was adjusted to 0.5 McFarland standard (1.5 × 10^8^ CFU/mL) turbidity. The 150 μL bacterial solution was transferred to a 96-well plate and aerobically cultured at 37 °C for 48 h. After the culture was completed, the contents of the wells were gently removed, and the wells were rinsed 3 times with 0.9% sterile 250 μL normal saline. After washing, 150 μL 0.1% crystal violet was added to each well for staining. After 15 min, the color was gently rinsed and dried in an oven at 60 °C, and then washed 3 times with 250 μL distilled water. Finally, 150 μL 96% ethanol was added. The optical density value at 570 nm was measured by quantitative spectrophotometry. According to the following criteria: (4ODc < OD570 = strong biofilm-forming bacteria, 2Odc < OD570 ≤ 4Odc = medium biofilm-forming bacteria, Odc < OD570 ≤ 2Odc = weak biofilm-forming bacteria, OD570 ≤ Odc = no biofilm-forming bacteria, Odc = negative control of the average OD570 + (3 × negative control standard deviation)), the isolates were divided into strong, medium, weak, or no biofilm-forming bacteria.

#### 2.2.5. Determination of Antibiotic Resistance

According to the agar dilution method of Lavalleé et al. [25], the prepared antibiotics with exponential concentration growth were added to the agar medium to make an antibiotic plate, and then the strain was inoculated to the surface of the medium. After 48 h of culture, the growth status of the strain was observed, the minimum inhibitory concentration (MIC) of the strain was determined, and the antibiotic resistance of the strain was judged. In this study, 6 categories of 9 antibiotics were selected as the test objects, as shown in Table 2.

### 2.3. Statistical Analysis

All data indicators are expressed as mean ± standard error of three replicates. Data were collated using Excel 2021, analyzed by one-way ANOVA using Graphpad Prism9.50, and separated by means using Duncan’s test. *p* < 0.05 indicates significance. The heat map was drawn using the Chiplot (https://www.chiplot.online, accessed on 28 November 2023) online tool.

## 3. Results

### 3.1. Bacterial Diversity of Seed-Borne Bacteria of A. splendens

A total of 41 strains of bacteria were isolated from the seeds of wild *A. splendens* from 6 regions. The number and species of bacteria isolated from various samples were different (Table 3).

#### 3.1.1. Taxonomic Status of Bacteria

According to the determination of the potassium hydroxide wire drawing test with the same nature of the Gram staining method, there were 17 strains of Gram-positive bacteria, accounting for 41.46%. There were 24 strains of Gram-negative bacteria, accounting for 58.54%. The 41 strains belonged to 3 phyla and 14 genera, of which 18 strains belonged to Firmicutes, accounting for 43.90%, 16 strains belonged to *Bacillus* sp., 1 strain belonged to *Lysinibacillus* sp., and 1 strain belonged to *Paenibacillus* sp. There were a total of twenty-two strains of Proteobacteria, accounting for 53.66%: two strains of *Klebsiella* sp., five strains of *Enterobacter* sp., one strain of *Atlantibacter* sp., five strains of *Pseudomonas* sp., one strain of *Brevundimonas* sp., three strains of *Siccibacter* sp., one strain of *Lelliottia* sp., two strains of *Erwinia* sp., one strain of *Mixta* sp., and one strain of *Stenotrophomonas* sp. There was one strain of bacteria belonging to Actinobacteria, accounting for 2.44%, and one strain of *Zhihengliuella* sp. (Figure 1).

#### 3.1.2. Bacterial Diversity Analysis of Seed-Borne Bacteria of *A. splendens*

The diversity of seed-borne bacteria of *A. splendens* was analyzed. Through a Wayne analysis, it was found that the common genus of bacteria isolated from six regions of Xinjiang, Inner Mongolia, Qinghai, Gansu, Ningxia Hui Autonomous Region and Jiangsu was Bacillus sp. (Figure 2A). The Circos species relationship diagram visually presents the information on the samples and the classification level of each bacterium, indicating the grouping information of the samples and the classification level of the bacterium, showing the percentage of the relative abundance of the bacterium in the samples. Different colors indicate the absolute abundance of bacteria (Figure 2B). As another presentation of the Wayne diagram, the UpSet diagram visually shows the common and unique parts among different regions. The UpSet diagram was used to analyze the composition of the bacterial genera in *A. splendens* in different regions. It was found that only *Bacillus* sp. was found in the six regions, and there were endemic genera in Xinjiang, Inner Mongolia, Ningxia and Gansu (Figure 2C). A random forest model was constructed with the genus of *A. splendens* as the independent variable, and the %IncMSE was used to score and rank the characteristics. The top three genera were *Bacillus* sp., *Enterobacter* sp., and *Siccibacter* sp., indicating that the species was more important for the identification of species in the group and that the bacteria of *Bacillus* sp., *Enterobacter* sp., and *Siccibacter* sp. were the marker species of the difference between groups (Figure 2D).

### 3.2. Mobility Analysis

By inserting the strain into the swimming medium by the acupuncture method, it was found that it could spread and grow on the surface of the plate, but its diffusion diameter was different (Figure 3). The results showed that the motion diameter of strains XJ16, GJ18, NM15, XNM19, NM6 and XJ3 on the surface of a semi-solid agar plate was close to the diameter of a culture dish 9 cm, and the mobility was strong, which was significantly different from other strains (*p* < 0.05), and the motion diameter conformed to the Wilson score interval, which is statistically significant. There were 20 strains of bacteria with a moving diameter of 2–8 cm, accounting for 48.78%. There were 17 strains of bacteria with a moving diameter of 0–2 cm, accounting for 41.46%, of which 7 strains had a moving diameter of less than 0.5 cm. The results showed that most of the bacteria carried by *A. splendens* could diffuse on the surface of the swimming medium, indicating that the bacteria carried by *A. splendens* had certain mobility and could migrate under suitable conditions. Because of this, bacteria with excellent performance could play a role in seeds.

### 3.3. Analysis of Biofilm Formation Ability

The biofilm formation ability was determined by crystal violet staining, ODC = 0.092. The results of the biofilm formation are shown in Figure 4. Strains XJ1, XJ2, XJ3, XNJ10, XNJ22, QJ1, XQJ4, GJ2, GJ6a, XGJ1a, XGJ4, XGJ8, XGJ12, NM8, XNM11, XNM15, JS-3, JS-25 and the other 18 strains were strong biofilm-forming bacteria. The 15 strains of XJ7, XJ12, XXJ6, XXJ9, XXJ17, XNJ16, QJ16, GJ15, GJ18, XGJ9a, NM6, NM10, NM15, JS-1 and JS-19 were medium biofilm-forming bacteria. The 11 strains of XJ16, QJ2, QJ14, XQJ2, NM27, NM31, NM42, XNM19, JS-4, JS-10 and JS-38 were weak biofilm-forming bacteria. Strain XXJ1 was a non-biofilm-forming bacterium. Among them, the proportion of bacteria with strong, medium, weak and no biofilm-formation ability was 45%, 37.5%, 27.5% and 2.5%, respectively.

### 3.4. Antibiotic Resistance Analysis

The agar dilution method was used to determine the bacterial resistance of *A. splendens*. The results showed (Figure 5) that the bacteria of *A. splendens* were more resistant to sulfadiazine, sulfamethoxazole and other sulfonamides, and the resistance to ampicillin and erythromycin was also more intuitive. Through the determination of the minimum inhibitory concentration of 41 strains of *A. splendens* seed-borne bacteria, it was found that the MIC of sulfadiazine was 2560 μg/mL, and the strains that could reach MIC were XJ1, XJ2, XJ3, XJ12, XXJ17, NM6, NM27, XNM19, XNJ10, XNJ22, QJ1, QJ2, XQJ4, XGJ4, JS-1 and JS-19. The MIC of sulfamethoxazole was 1280 μg/mL, and the strains that could reach MIC were XXJ1, XXJ6, QJ14, GJ18, XGJ1a, JS-4 and JS-38. The MIC of ampicillin was 1280 μg/mL; only XXJ17 could reach MIC. The MIC of erythromycin was 1280 μg/mL; only XJ1 could reach MIC. The MIC of kanamycin was up to 10 μg/mL. The bacteria that could reach MIC were XJ1, XJ2, XJ3, XJ12, XXJ17, NM27, XNM19, QJ2, QJ14, JS-4 and JS-19. The MIC of rifampicin was up to 80 μg/mL, and only XXJ17 could reach MIC. The MIC of gentamicin was up to 20 μg/mL, and only QJ14 could reach MIC. The MIC of ceftazidime was up to 640 μg/mL, and the bacteria that could reach MIC were NM10, XGJ1a and XGJ9a.

### 3.5. Correlation Analysis

Through the analysis of bacterial motility, biofilm formation ability and antibiotic resistance of *A. splendens*, as shown in Figure 6, the results showed that there was no correlation between bacterial motility and biofilm formation ability and antibiotic resistance. The biofilm formation ability was significantly correlated with the resistance of bacteria to aminoglycosides such as kanamycin, amikacin and gentamicin (*p* < 0.01); the correlation analysis was statistically significant. And there was no correlation between antibiotic resistance and motility and biofilm formation ability.

## 4. Discussion

In this study, 41 dominant bacteria were selected from 272 species-borne bacteria isolated from *A. splendens* in different regions for preliminary study. The analysis showed that the 41 strains belonged to 3 phyla and 14 genera, mainly Proteobacteria and Firmicutes, followed by Actinobacteria, which was consistent with previous research results. Studies have shown that 131 genera of natural endophytic bacteria have been identified in the seeds of 25 common plants. Most of the endophytic bacteria in seeds belonged to Proteobacteria, among which Enterobacteriaceae are the dominant endophytic bacteria in a variety of plant seeds [26,27]. In addition, Actinobacteriota, Firmicutes and Bacteroidetes were also found in wheat seeds [28]. In the past two decades, many researchers have found that Gramineae and Leguminosae seeds carry bacteria, among which Proteobacteria, Actinobacteria and Firmicutes are the most abundant [29,30]. At the same time, through the molecular identification of the 41 strains in this experiment, it was found that the bacteria carried by the seeds in different regions were significantly different in species and quantity, which may be related to the storage life of the seeds and the ecological environment of the year. *Bacillus* was the dominant genus among the 41 strains, which was detected in the 6 regions of Xinjiang Uygur Autonomous Region, Inner Mongolia Autonomous Region, Ningxia Hui Autonomous Region, Qinghai Province, Gansu Province and Jiangsu Province. It has been reported that Bacillus is also present in other plant seeds as a dominant bacteria. In the study of different seeds, such as *Medicago sativa* [31], *Trifolium repens* L. [27], *white clover* [32] etc., it was found that the microorganisms carried by the seeds were diverse. This conclusion is consistent with the results of previous reports Truyens [27]. For the study of *Achnatherum inebrians* [33], *Elymus sibiricus* and *Elymus dahuricus* [32], *Avena sativa* L. [34], etc., it was found that the dominant bacteria of most plants were Bacillus. Through the study and analysis of culturable bacteria in the *Achnatherum splendens* seed belt, it was found that they also had diversity. The phenotype of most plants is the product of synergistic and highly co-regulated expression of plant genes and microbial genes.

The close connection between the seed-borne bacteria and the seed causes a specific co-evolutionary effect, and the bacteria that mutually benefit from the seed also continue due to the vertical transmission of the seed-borne bacteria [35]. Studies have shown that the bacteria carried by seeds can not only be transmitted across generations by vertical transmission, but can also be transmitted through vascular bundles from parent plants into seeds [36]. This mode of transmission of bacteria ensures that they can exist in the natural environment for a long time without being eliminated. In addition, studies have found that under the influence of biotic and abiotic stresses, the presence of seed-borne bacteria can improve the viability, germination and resilience of seeds, improve the growth and development of plants, and thus enhance the resistance of plants to adverse effects [37,38]. In this paper, through the study of bacterial resistance of *A. splendens*, it was found that the bacteria carried by *A. splendens* seeds had different degrees of resistance to different kinds of antibiotics. The presence of bacteria in the seed band could alleviate the stress of seeds in the antibiotic environment to a certain extent, which laid a foundation for the germination and growth of seeds in the antibiotic environment.

Bacteria carried by different seeds have different mechanisms of action on plants. According to the reports of Xu [4] and Walitang et al. [39] seed-borne bacteria can directly or indirectly promote the growth of host seeds and seedlings by producing hydrolases, regulating hormones and other abilities, and their own growth-promoting characteristics such as nitrogen fixation, phosphorus solubilization and siderophore production. Compared with the studies on the growth-promoting functions of seed-borne bacteria and plant growth-promoting rhizobacteria (PGPR), such as nitrogen fixation, phosphorus solubilization and hydrolase production, there are few reports on the characteristics of seed-borne bacteria [40]. Seed-borne bacteria can not only affect the germination and growth of seeds and even plants by their unique functions, but also affect the host due to their potential uses. It has been reported that there is a significant correlation between undiscovered microbial diversity and potential uses. Smith et al. [41] found that newly discovered bacteria can produce bioactive substances through the investigation of endophytic bacteria in tropical plants, which has potential significance in the fields of medicine, agriculture and industry. In 2013, Shi et al. [32] studied the insecticidal activity of endophytic bacteria from *A. splendens*, and found that the isolates had good insecticidal activity against cotton aphids, which could be used to develop new natural insecticides. In this study, the characteristics of motility, biofilm formation and antibiotic resistance of 41 strains isolated from the seeds of *A. splendens* were qualitatively or quantitatively studied. It was found that the motility, biofilm formation and antibiotic resistance of each bacteria were different. In the future, more in-depth studies will be carried out on strains with excellent characteristics to determine whether they have the ability to promote growth, prevent disease, and their effect on plants. Future studies will be committed to applying it to seed coating, microbial fertilizer and other aspects by virtue of its growth-promoting performance, and applying it to new species breeding, environmental pollution control and plant disease prevention by virtue of its ability to resist stress and prevent disease.

Motility is one of the important characteristics of bacterial physiology, and plays a vital role in microbial ecology. The motility of bacteria enables them to establish contact with animals and plants. In this study, the motility of *A. splendens* seed-borne bacteria on the surface of a swimming medium containing 0.3% agar was investigated. It was found that six strains, NM6, XNM19, NM27, NM15, GJ18 and XJ16, had strong motility. They belonged to *Siccibacter* sp., *Enterobacter* sp., *Atlantibacter* sp., and *Lelliottia* sp., and their diffusion area almost covered the entire surface of the medium. We speculate that this may be because the nutritional conditions of the swimming medium were suitable for the movement and diffusion of these six strains. It may also be that these six strains have flagella tissue that can promote their diffusion. Study showed that if the nutritional conditions of the parasitic environment are good, the bacterial motility will be stronger, and the colonization area covered by their migration will be larger [42]. In addition, exercise can improve the efficiency of nutrient acquisition and avoid toxicity [43]. It has been reported that motility enables bacteria to migrate to favorable environments and stay away from the effects of toxic substances and their predators [44,45]. Samad et al. [42] found that although the motility of Staphylococcus bacteria is limited, it can still migrate with other mobile bacteria as carriers, and reach and survive and reproduce in its preferred niche, indicating that non-mobile bacteria can migrate under the influence of mobile bacteria. In addition, studies have shown that flagella-mediated bacteria not only have strong motility, but also participate in the formation of bacterial biofilms due to the physical adsorption of flagella. As early as 1987, Lawrence et al. [46] demonstrated that flagella, motility, and chemotaxis play an important role in the formation of bacterial biofilms; bacterial motility makes it spread to the appropriate niche to adhere and proliferate, thus promoting the formation of biofilm [47].

In addition, studies have shown that biofilms can make bacteria better adapt to the living environment, especially strains with poor antibiotic resistance, and even if the biofilm formation ability of bacteria with high antibiotic resistance is weak, it can still protect strains from persecution [48]. However, in 1998, George A. et al. [49] found that the adhesion ability of flagellar-defective mutants on PVC was very poor when studying the formation mechanism of Pseudomonas aeruginosa biofilm, indicating that flagella was indispensable in the process of biofilm formation and that bacterial motility was positively correlated with its biofilm formation ability. Their research showed that there was a positive correlation between bacterial motility, biofilm formation ability, and antibiotic resistance. It has also been reported that bacterial motility is negatively correlated with biofilm formation. Kim et al. (2008) [50] explored the effect of flagellar mutations on the biofilm formation of *Y. enterocolitica*. It was found that for the virus, if the gene that is a positive regulator of flagellar expression mutates, the biofilm formation will decrease, while the gene mutation that encodes the negative regulator of flagellar expression will make the biofilm formation stronger. In this study, the resistance of *A. splendens* to sulfonamide antibiotics was stronger than that of other antibiotics. Among them, the number of *Bacillus* was the most resistant, followed by the bacteria of *Pseudomonas* sp., *Enterobacter* sp. and *Siccibacter* sp., *Pseudomonas* sp., and *Bacillus* can enhance the ability of hosts to resist pathogens by producing bioactive substances [51]. There was a significant positive correlation between biofilm formation ability and aminoglycoside antibiotic resistance (Figure 6). The reason may be that the presence of biofilms can provide an advantage for bacteria to better exist in antibiotic-contaminated environments. Their co-occurrence and correlation indicate that they may have a common regulatory mechanism, which may also be the reason why bacteria initiate biofilm gene expression. There was no significant correlation between motility and biofilm formation ability, and there was no significant correlation with bacterial resistance. The reason may be that the number of variables was too small to make the data results deviate from the expected, or the number of repeated values of a single strain was too small to analyze separately and the relationship between the two variables was not observed.

The bacterial diversity of *A. splendens* from different regions is different, but most strains have certain motility and biofilm formation ability, and can produce resistance to one or more antibiotics. The strains with excellent biological characteristics were screened out, which provided fungal resources for improving the tolerance of plants under different stresses.

## 5. Conclusions

(1) The culturable bacteria in the seed-borne bacteria of *Achnatherum splendens* have species diversity. The isolated bacteria are identified to belong to 3 phyla and 14 genera, among which Bacillus is the dominant genus in the seed zone of *Achnatherum splendens* in different regions.

(2) In total, 41 strains of bacteria isolated from *Achnatherum splendens* are motile, and 25 strains of bacteria have a moving diameter of >2.00 cm.

(3) Among the 41 strains, 16 strains are strong biofilm-forming bacteria, 14 strains are medium biofilm-forming bacteria, 10 strains are weak biofilm-forming bacteria, and 1 strain has no biofilm-forming ability.

(4) The bacteria carried by *Achnatherum splendens* have certain resistance to macrolides (erythromycin), cephalosporins (ceftazidime), penicillins (ampicillin), aminoglycosides (kanamycin, amikacin, gentamicin), and sulfonamides (sulfadiazine, sulfamethoxazole), among which the resistance to sulfonamides was the strongest.

## Figures and Tables

**Figure 1 microorganisms-12-00339-f001:**
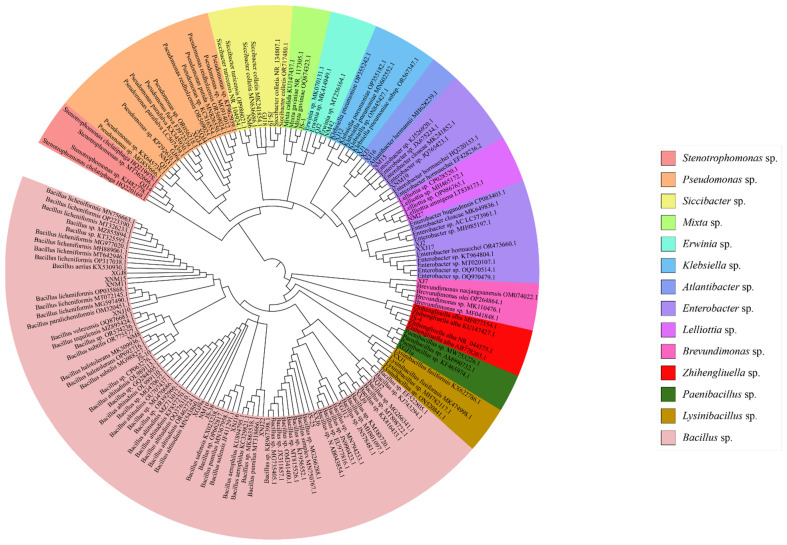
Phylogenetic tree of the species of seed-borne bacteria of *Achnatherum splendens*.

**Figure 2 microorganisms-12-00339-f002:**
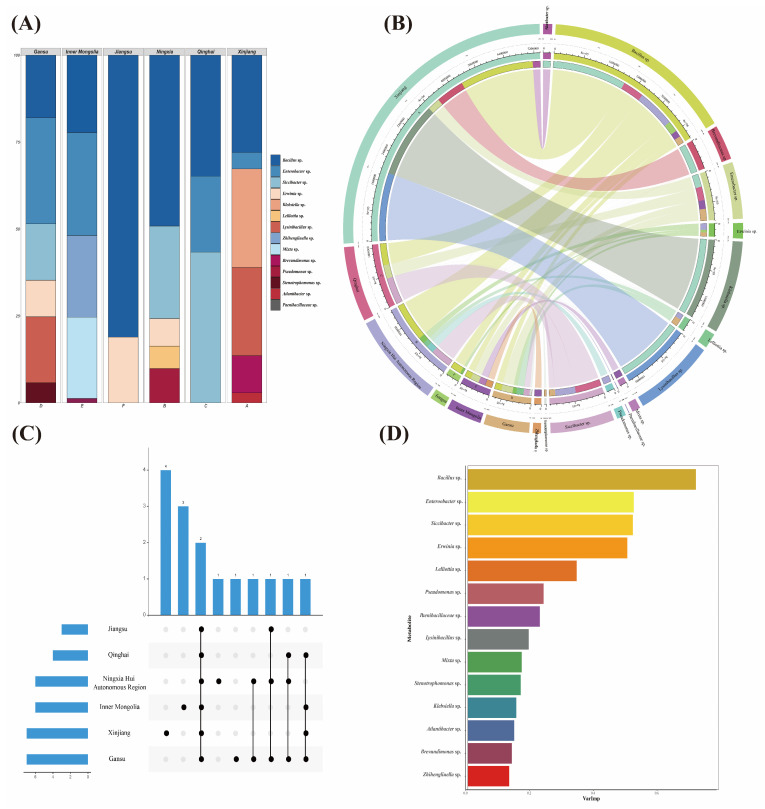
Analysis of bacterial diversity of seed-borne bacteria of *Achnatherum splendens.* (**A**) Describes the classification and proportion of bacteria contained in each region; (**B**) describes the bacteria contained in each region and the relationship between regions and bacteria; (**C**) intuitively describes the common and unique genera among bacteria in each region; (**D**) describes the importance of species-borne bacteria in the host.

**Figure 3 microorganisms-12-00339-f003:**
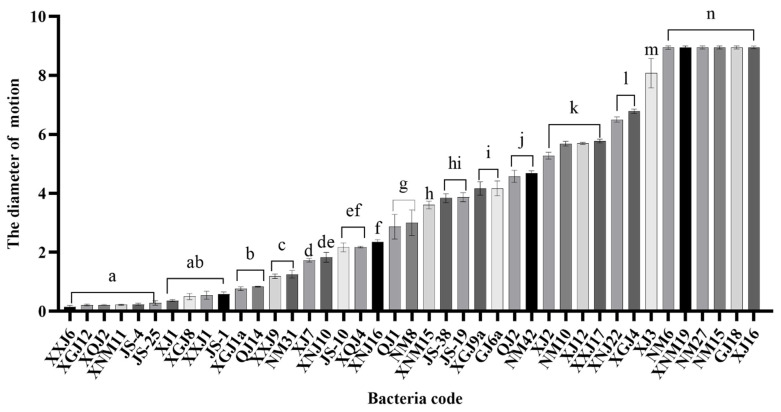
Analysis of bacterial motility of seed-borne bacteria of *Achnatherum splendens*. Different letters on the column indicate the significant difference level of bacterial motility.

**Figure 4 microorganisms-12-00339-f004:**
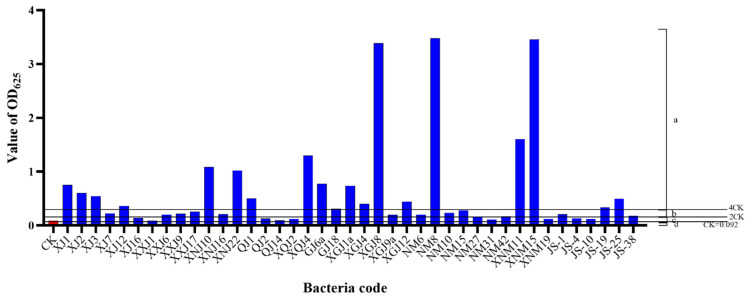
The biofilm formation capacity of the *Achnatherum splendens* with seed-borne bacteria. Note: a is a strong biofilm formation ability; b is moderate biofilm formation ability; c is weak biofilm formation ability; d is no biofilm formation ability. The red column represents the value of the control group, and the blue is the value of the experimental group.

**Figure 5 microorganisms-12-00339-f005:**
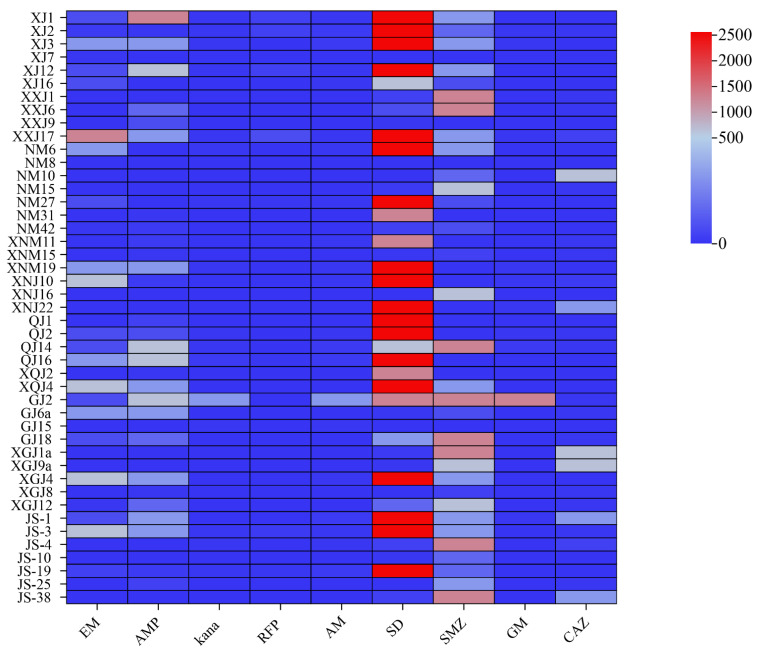
The antimicrobial resistance of the *Achnatherum splendens* with seed-borne bacteria.

**Figure 6 microorganisms-12-00339-f006:**
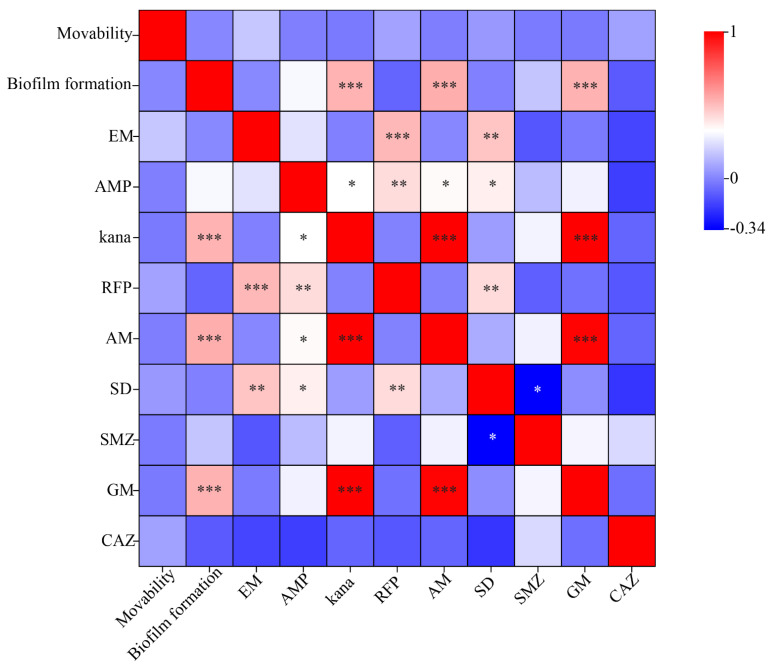
Correlation analysis of biological characteristics. Note: |r| < 0.3 is low correlation, 0.3–0.8 is moderate correlation, |r| > 0.8 is highly correlated. * means if *p* < 0.05, ** means if *p* < 0.01, *** means if *p* < 0.001, there is statistical significance.

**Table 1 microorganisms-12-00339-t001:** Related information of *Achnatherum splendens* species.

Origin	Grass Species	Production Time	Source
Xinjiang Uygur Autonomous Region	*Achnatherum splendens*	2020	Seeds were purchased from Qinghai Womiao Ecological Technology Co., Ltd., Xining, China
Inner Mongolia Autonomous Region	*Achnatherum splendens*	2021	
Qinghai Province	*Achnatherum splendens*	2022	
Gansu Province	*Achnatherum splendens*	2021	
Ningxia Hui Autonomous Region	*Achnatherum splendens*	2021	
Jiangsu Province	*Achnatherum splendens*	2020	

**Table 2 microorganisms-12-00339-t002:** Antibiotics used for antibiotic resistance determination and test concentration.

Antibiotic Name	Classification of Antibiotics	Screening Concentration (µg/mL)
EM	Macrolides antibiotics	10
ANP	β-lactams	20
Kana	Aminoglycosides	40
CAZ	Cephalosporins	80
GM	Aminoglycosides	160
SD	Sulfonamides	320
SMZ	Sulfonamides	640
AM	Aminoglycosides	1280
RFP	Rifamycins	10, 20, 40, 80, 160, 320, 640, 1280, 2560

**Table 3 microorganisms-12-00339-t003:** Identification results and morphological description of seed-borne bacteria of *Achnatherum splendens*.

Strain Code	Gram	16S rDNA Identification	Accession Number
XJ1	−	*Klebsiella* sp.	OR889436
XJ2	−	*Enterobacter* sp.	OR889437
XJ3	−	*Atlantibacter* sp.	OR889438
XJ7	−	*Brevundimonas* sp.	OR889439
XJ12	−	*Klebsiella* sp.	OR889440
XJ16	−	*Enterobacter* sp.	OR889441
XXJ1	−	*Lysinibacillus* sp.	OR889445
XXJ6	+	*Bacillus* sp.	OR889446
XXJ9	+	*Bacillus* sp.	OR889447
XXJ17	−	*Enterobacter* sp.	OR889448
XNJ10	+	*Bacillus* sp.	OR889455
XNJ16	+	*Bacillus* sp.	OR889456
XNJ22	+	*Bacillus* sp.	OR889466
QJ1	−	*Pseudomonas* sp.	OR889452
QJ2	−	*Erwinia* sp.	OR889453
XQJ2	+	*Bacillus* sp.	OR915404
QJ14	−	*Stenotrophomonas* sp.	OR889451
XQJ4	−	*Pseudomonas* sp.	OR889467
GJ6a	−	*Pseudomonas* sp.	OR889449
GJ18	−	*Siccibacter* sp.	OR889450
XGJ1a	+	*Paenibacillus* sp.	OR889463
XGJ9a	+	*Bacillus* sp.	OR889468
XGJ4	−	*Pseudomonas* sp.	OR889464
XGJ8	+	*Bacillus* sp.	OR889454
XGJ12	+	*Bacillus* sp.	OR889465
NM6	−	*Siccibacter* sp.	OR889429
NM8	+	*Bacillus* sp.	OR889430
NM10	+	*Bacillus* sp.	OR889431
NM15	−	*Enterobacter* sp.	OR889432
NM27	−	*Lelliottia* sp.	OR889433
NM31	−	*Pseudomonas* sp.	OR889434
NM42	−	*Erwinia* sp.	OR889435
XNM11	+	*Bacillus* sp.	OR889442
XNM15	+	*Bacillus* sp.	OR889443
XNM19	−	*Enterobacter* sp.	OR889444
JS-1	−	*Mixta* sp.	OR889457
JS-4	+	*Zhihengliuella* sp.	OR889458
JS-10	+	*Bacillus* sp.	OR889459
JS-19	−	*Siccibacter* sp.	OR889460
JS-25	+	*Bacillus* sp.	OR889461
JS-38	+	*Bacillus* sp.	OR889462

Note: +: Gram-positive bacteria; −: Gram-negative bacteria.

## Data Availability

Data are contained within the article.

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
