# Peer review of "Diversity and Biological Characteristics of Seed-Borne Bacteria of Achnatherum splendens"

_microorganisms, 2024, doi:10.3390/microorganisms12020339_

Round 1
Reviewer 1 Report
Comments and Suggestions for Authors
Introduction
What I found weakly supported is the problem presented.
Seed-borne bacteria is a normal mechanism that bacteria use to survive, as pointed out in introduction, the connection with the model plant and antibiotic resistance does not match easily. It feels like forcing something to enter in an improper part.
So, the introduction must be rewritten, especially the description of the problem.
Consider that the information provided in introduction may help to understand the importance of the study.
See comments in MS.

Reviewer 2 Report
Comments and Suggestions for Authors
The paper submitted to me for review, entitled "Diversity and Biological Characteristics of Seed-borne Bacteria of Achnatherum splendens", was very well and carefully prepared and certainly deserves to be published in the journal Microorganisms. Nevertheless, I have taken the liberty of including a few suggestions for the authors below, which I think could enrich the evaluated paper somewhat:
Materials and methods
DNA extraction and molecular identification:
- Provide more details about the PCR amplification procedure, such as the duration of each step, the annealing temperature and the number of cycles.
Determination of antibiotic resistance:
- Provide details of the antibiotics used in the study. Please indicate the range of concentrations tested and explain the choice of these specific antibiotics.
Statistical analysis:
- Provide a brief description of the Chiplot platform and its role in generating a heat map.
Result
Statistical significance:
- Statistical analyses should clearly state whether differences or correlations are statistically significant and the level of significance (e.g., p < 0.05).
Discussion
- Begin the discussion with a brief summary of the main aims of the study and a concise overview of the results.
- Explain the significance of the study in the broader context of research on seed bacteria.
- Clearly state how the results are consistent with or different from previous research mentioned in the discussion. Provide specific references to support these comparisons.
- Discuss potential applications of the discoveries in agriculture, industry and microbial biotechnology. Explore how investigated strains with excellent properties can be used in practical applications, e.g. to prevent plant diseases or to develop microbial fertilisers.
Reviewer 3 Report
Comments and Suggestions for Authors
The article Diversity and Biological Characteristics of Seed-borne Bacteria of Achnatherum splendens analyzed bacterial isolates from the seeds of A. splendens in six different provinces of China. Some strains showed strong biofilm formation ability. They found that sulfonamides had the strongest antibacterial ability in strains. As that study provides fungal resources for improving the tolerance of plants under different stresses, it is the first report on the biological characteristics of bacteria in A. splendens. Therefore, I recommend the manuscript for publication, but I have some comments to improve it.
Abstract: No comments.
Introduction:
Lines 43-44: ….from scholars?…It is not fine; please rewrite these lines.
Line 45: Please avoid using the word scholars; it is a secondary term.
Lines 53 – 81: That paragraph is too long; please break it down into 2 or more paragraphs.
Other comments: I would recommend to state the hypothesis.
Material and Methods:
Please describe more in section 2.1. Experimental materials, it has only a table and no text.
Line 106: It needs an English proofreading. The following sentence is not clear:
Add 10 mL sterile water to the disinfected and unsterilized seeds for full…
It seems that this line was copied and pasted from some protocol.
2.3. Statistical Analysis:
It is necessary if the ANOVA’s assumptions have been attempted. Please double-check if the data have normality and homoscedasticity.
Results:
The results are well written.
Lines 185-205 and Lines 213-228: Paragraphs are too long.
Discussion:
1) Please rewrite the beginning of the discussion; the current way is a literature review. It is necessary to show the most important results and explain their reasons.
2) All paragraphs of the discussion are too long.
Conclusions:
Please state the conclusions in the present tense.
